# Health Priorities and Participation in Peer-Led Active Rehabilitation Camps Among Persons with Spinal Cord Injury: A Prospective Cohort Study

**DOI:** 10.3390/jcm15010176

**Published:** 2025-12-25

**Authors:** Tomasz Tasiemski, Piotr Kazimierz Urbański, Dawid Feder, Magdalena Lewandowska, Anestis Divanoglou

**Affiliations:** 1Department of Adapted Physical Activity, Faculty of Sport Sciences, Poznan University of Physical Education, 61-871 Poznań, Poland; urbanski@awf.poznan.pl; 2Foundation for Active Rehabilitation, 02-574 Warsaw, Poland; dawid.feder@far.org.pl; 3Calculation Centre, Poznan University of Physical Education, 61-871 Poznań, Poland; lewandowska@awf.poznan.pl; 4Clinical Department of Pain and Rehabilitation Medicine in Linköping, Region Östergötland, SE-58183 Linköping, Sweden; anestis.divanoglou@liu.se; 5Department of Health, Medicine, and Caring Sciences, Linköping University, SE-58183 Linköping, Sweden

**Keywords:** health priorities, spinal cord injury, peer-led Active Rehabilitation Camps, Inter-PEER project

## Abstract

**Background**: Peer-led Active Rehabilitation Camps (ARC) aim to enhance functional independence and self-management among people with spinal cord injury (SCI). In Poland, where access to specialized spinal units and lifelong follow-up is limited, these programs may help address key health priorities—mobility, bowel and bladder management, sexual well-being, and upper-limb function. This study examined whether participation in ARC helped individuals achieve these priorities and identified factors associated with outcomes. **Methods**: This prospective cohort study, part of the Inter-PEER project, included 125 adults with SCI who attended one of 16 consecutive ARCs in Poland (2023–2024). Eligible participants used a manual wheelchair, were aged ≥ 16 years, and could complete written questionnaires. Data were collected at camp start (T1), completion (T2), and 3-month follow-up (T3) using surveys and wheelchair skills assessments. Validated instruments (SCIM-SR, MSES, QEWS, WST-Q, LiSat-11) were used and were aligned with the four priority domains. Associations with demographic and injury variables were examined using multivariate regression analyses. **Results**: Participants showed significant gains across priorities during the 10-day ARC. Mobility improved on all wheelchair-skill measures (e.g., QEWS + 2.6 points, *p* < 0.001), with most gains sustained at T3. Among participants with tetraplegia, self-care and hygiene scores increased by 24% and remained elevated at follow-up. Confidence in achieving a satisfying sexual relationship increased by camp end and was accompanied by higher sexual-life satisfaction at T3. Regression analyses found only modest associations between outcomes and demographic or injury characteristics. **Conclusions**: Participation in peer-led ARC programs was associated with rapid, clinically meaningful improvements in several health domains prioritized by people with SCI, especially upper-limb function, sexual well-being, and wheelchair mobility. Our findings highlight the value of integrating structured, peer-based community programs into the continuum of SCI rehabilitation.

## 1. Introduction

Spinal cord injury (SCI) results in permanent disability, functional limitations, and lifelong healthcare needs, making it one of the most burdensome conditions in terms of disability-adjusted life years in Western Europe [1]. Community dwelling individuals with SCI indicate that their top health priorities are restoring mobility, bowel, bladder, and sexual function [2,3,4]. Across international datasets, satisfaction with sexual life remains the lowest-rated area of life satisfaction in this population [5,6]. For individuals with tetraplegia, regaining arm and hand function is also a key priority [2]. These health priorities remain consistent regardless of time since injury [2,3,4]. Addressing them requires a coordinated rehabilitation continuum, encompassing both hospital-based and community-based services [7,8].

In Poland (population 38 million), rehabilitation of persons with SCI is largely provided in general hospitals rather than specialized spinal units—only two of which exist nationally [9]. Lifelong follow-up is inconsistent and varies across regions [9]. The Foundation for Active Rehabilitation (FAR) is the main provider of community-based interventions for individuals with SCI [8,9]. Its core activity, the peer-led Active Rehabilitation Camp (ARC), aims to enhance independence and self-management through structured physical training and education [8,10]. Given the sparsity of specialized services, ARCs may fill critical gaps in long-term rehabilitation for people with SCI in Poland.

Emerging international evidence suggests that ARCs can have positive effects in various outcome areas for individuals with SCI [11,12]. A recent Swedish study reported significant gains in physical independence—particularly in dressing, washing, bowel management, transfers, and wheelchair skills—maintained at three months [13]. Improvements in self-efficacy and resilience observed at ARC completion were not sustained at the 3-month follow-up [13]. At a similar camp in Botswana, participants with SCI achieved large effects in mobility and wheelchair skills, and moderate effects in disease management self-efficacy, with gains maintained at 5 months post-program [11]. In Poland, large improvements in wheelchair skills were reported at ARC completion and retained at 3-month follow-up [12]. While these findings are promising, it remains unclear whether ARCs address the specific health priorities identified by people with SCI and which factors are associated with greater improvements.

This study aimed to (1) assess the extent to which participation in ARC in Poland helps individuals with SCI fulfill key health priorities, and (2) identify factors associated with these outcomes. We hypothesized that, despite the short duration of the 10-day intervention, participation in ARC would lead to improvements in wheelchair skills and functional independence, with gains largely maintained at the 3-month follow-up.

## 2. Materials and Methods

### 2.1. Design

This prospective cohort study is part of the Inter-PEER project and follows its published protocol [10]. The Inter-PEER protocol presents a systematic evaluation of the effects of ARCs among individuals with SCI using standardised outcome measures aligned with the camp objectives [10]. Ethical approval was obtained by the Bioethics Committee at the Medical University in Poznań, Poland.

### 2.2. Setting and Participants

The study included 16 consecutive ARCs conducted in Poland between May 2023 and February 2024 with study participants representing different regions of the country. All participants of ARC were invited to participate in the study if they met the following inclusion criteria: (1) having a SCI (acquired traumatic and non-traumatic, and congenital, e.g., spina bifida); (2) being 16 years or older; (3) being able to comprehend and answer written questions. All participants received written and verbal information about Inter-PEER, and provided their written informed consent before enrollment. Participation in the study was voluntary and anyone could withdraw at any time without specifying the reason.

### 2.3. Procedures

The ARC, previously described using the Template for Intervention Description and Replication TIDieR framework [10], is a short (average 10 days), intensive (about 8 h per day), group-based, peer-led training program conducted in community settings. It includes training sessions, educational workshops, and informal interactions with peer mentors—individuals living with SCI. The program focuses on training activities of daily living (ADL), wheelchair mobility, and self-management as well as educational sessions address prevention of secondary complications such as urinary tract infections, pressure sores, bowel management, sexuality, fertility, and parenting [8,10]. In Poland participants also receive at least 12 h of face-to-face training in their home environment before and after the camp to reinforce physical independence [12].

Participants were assessed at three time points: camp start (T1), completion (T2), and 3-month follow-up (T3). Questionnaires were administered through online surveys using project-provided tablets at T1 and T2, with an on-site coordinator available to clarify queries. At these study points, participants also completed practical wheelchair skills assessments administered by peer mentors. At T3, participants received an individualized survey link to complete remotely.

Program fidelity—the extent to which ARCs adhered to the standardized Inter-PEER protocol—was monitored for each camp [10]. Program fidelity was high. The peer mentor-to-participant ratio of 1:2–3, with mentors leading an average of 80% of all sessions. Daily schedules consisted of 8.6 training hours (each 45 min), comprising 3.3 h of ADL and wheelchair skills training, 3.4 h of physical, sports, and recreational activities, and 1.9 h of formal education. Each ARC lasted 10 days and typically involved nine participants and four peer mentors with SCI. Additional fidelity criteria are provided in Appendix A.

### 2.4. Outcomes

For this study, all Inter-PEER variables were reviewed and matched to the relevant health priorities using the highest (more overarching) applicable level of scoring: (a) when the total score was fully relevant, it was used; (b) when the total score included some irrelevant items, the appropriate domain score was used; (c) when a domain still included irrelevant items, only the relevant individual item was used.

Mobility function (walking & wheeling) was matched with four Inter-PEER outcome measures:
➢The Queensland Evaluation of Wheelchair Skills (QEWS), an SCI-specific measure that assesses the abilities of: (1) negotiating an indoor circuit; (2) ascending and descending a ramp; (3) maintaining balance on the back wheels; (4) ascending and descending a gutter; and (5) distance covered during a six-minute push test [14]. Each task is scored 0–5 (total: 0–25). QEWS is simple to administer, reliable, valid and sufficiently sensitive for detecting changes during ARC and a 10-week period of inpatient rehabilitation [11,14]. The internal consistency of the QEWS was satisfactory in this study cohort (α = 0.84–0.86).➢The Wheelchair Skills Test Questionnaire (WST-Q, v4.3) for manually operated wheelchairs was used to subjectively assess capacity and confidence in 24 wheelchair skills, with capacity and confidence scores converted to a 0–100 scale [15]. The WST-Q has demonstrated strong content, construct, and concurrent validity for individuals with SCI [15]. A success rate for individual skills ≥ 20% was considered clinically significant [15]. The internal consistency for the WST-Q capacity in this study cohort was satisfactory for particular study terms (α = 0.95–0.96) and for the WST-Q confidence (α = 0.95–0.96).➢The mobility indoors/outdoors domain of the Spinal Cord Independence Measure Self-report (SCIM-SR) [16]. The SCIM-SR evaluates the level of independence in individuals with SCI based on 17 items divided into three domains: (1) self-care (eating, grooming, bathing, dressing), (2) respiration and sphincter management, and (3) mobility (in room/toilet and indoors/outdoors). Domain scores range between 0 and 40 with higher scores indicating a higher functional level. Each item is weighted according to the subjective value of the activity, the difficulty level of performing the task, and the time required. The internal consistency for the SCIM-SR domains in this study cohort was satisfactory (α = 0.78–0.80).➢The corresponding item of the Moorong Self-efficacy scale (MSES), i.e., Get out of my house (Item 5) [17]. The MSES is a 16-item scale rating confidence in the ability to control behaviour and outcomes on a 7-point Likert scale (1 = very uncertain, 7 = very certain) with higher scores indicating high self-efficacy [18]. It was developed specifically for people with SCI, and comprises three domains: (1) personal function, (2) social function, and (3) general self-efficacy [18]. The MSES has shown strong evidence of construct validity, stability and internal consistency [17]. The internal consistency for the MSES in this study cohort was satisfactory for particular study terms (α = 0.88–0.91).
Bowel and bladder functions were matched with two subjective measures:➢The corresponding items of the SCIM-SR, i.e., Bladder management (Item 6), Bowel management (Item 7), Using the toilet (Item 8), Transfer from the wheelchair to the toilet/tub (Item 11) [16].➢The corresponding item 2 of the MSES, i.e., I can avoid having bowel accidents.
Sexual function was matched with two Inter-PEER outcome measures:
➢The corresponding item of the Life Satisfaction Questionnaire (LiSat-11), i.e., satisfaction with sexual life (Item 5) [18]. The LiSat-11 consists of 11 items covering global life satisfaction (one item) and domain-specific satisfaction across ten areas. Each item is rated on a 6-point scale from 1 (very dissatisfied) to 6 (very satisfied), with higher scores indicating greater life satisfaction. The LiSat-11 is valid for the general population [18] and has demonstrated satisfactory internal consistency in individuals with SCI [19]. In this study cohort, the internal consistency of the LiSat-11 was satisfactory (α = 0.85–0.86).➢The corresponding item 6 of the MSES, i.e., I can have a satisfying sexual relationship.
Management of the tetraplegic upper limb aims to maximize hand function to enable performance of daily tasks as independently as possible [20]. Based on that, hand and arm function in individuals with tetraplegia was matched with three Inter-PEER outcome measures:
➢The self-care SCIM-SR domain (α = 0.90–0.93) [16].➢The corresponding item of the MSES, i.e., Maintain my personal hygiene (Item 1).➢The item 7 of the LiSat-11, i.e., My ability to manage my self-care (dressing, hygiene, transfers, etc.) [18].


Data on sociodemographic and injury-related factors were collected using 17 questions adapted from the International Spinal Cord Injury Community Survey [21].

### 2.5. Statistical Analysis

Statistical analysis employed descriptive statistics: mean, standard deviation (SD), frequency (n), percent (%), interquartile range (IQR) and 95% confidence interval (95%CI). Normality was assessed using histograms and the Kolmogorov-Smirnov test. As data were not normally distributed, the Wilcoxon test was used for two measurement points (T1/T2 and T1/T3). Effect sizes (d) were calculated as mean difference/SD of the difference and categorized per Cohen’s criteria: small (≥0.2–<0.5), moderate (≥0.5–<0.8), and large (≥0.8). The sample required to detect changes in the small-to-moderate effect size for this study was established at 101 participants [10]. We identified and reported individuals who reached a minimum clinically important difference (MCID) using the formula 0.2 × SDstart [22]. The MCIDs for the domain and total scores for all primary Inter-PEER outcome measures are available based on the Swedish population of persons with SCI attending ARC (QEWS ≥ 1.0; WST-Q capacity and confidence ≥ 5.0; Mobility indoors/outdoors SCIM-SR domain ≥ 1.0; Self-care SCIM-SR domain ≥ 1.0) [13]. Our study confirmed the above MCIDs based on persons with SCI attending ARC in Poland with exception for WST-Q confidence ≥ 6.0.

To explore associations between demographic and injury factors and functional gains, four health priority indices (dependent variables) were developed by integrating data from matching outcome measures. Outcome measures from the Inter-PEER project, or their relevant elements (domains or individual items), were grouped into four composite indices based on semantic consistency, reflecting the same health priority. These indices are exploratory composite constructs and not validated outcome measures. A sensitivity analysis was performed to evaluate the robustness of the pooled effect estimates. When individual components were sequentially included or excluded from the composite indices, we examined their influence on the pooled mean difference (D). For the Mobility Index, inclusion of MSES item 5 (“Get out of my house”) substantially altered the pooled effect estimate. Similarly, for the Bowel and Bladder Index, SCIM item 6 and MSES item 2 attenuated the pooled effect. Therefore, each composite index was recalculated repeatedly with single-component removal. In all cases, the pooled effects remained statistically significant (*p* < 0.05), and the magnitude of D remained stable across all four indices. For each index, only participants with complete data were included, and components showing significant change between time points were retained. Since each of the components had a different score range, we normalized their results to percentage values (range from −100% to 100%) according to the mathematical formula:$$Pn\left[\%\right]=\frac{R_{T2}-R_{T1}}{D}\times 100\%$$

Pn [%]—result expressed as a percentage of each component.

R_Tn_—scores of the examined person in the term T1/T2 or T1/T3.

D—maximum score difference possible to obtain in a given scale.

Meta-analysis and meta-regression modules in Statistica software version 13 (StatSoft Polska Sp. z o.o. 2025; version 5.1.0.; www.statsoft.pl) were used to analyze paired mean differences and estimate component contributions. The contribution values were calculated taking into account the strength of correlation between measurements, sample size, and the statistical significance of the observed differences. Four health priority indices were created separately for T1/T2 and T1/T3. The resulting summary effect for each index served as the dependent variable in subsequent regression analyses.

All data underwent screening for regression analysis suitability. The Mann-Whitney U-test (Z), *t*-test and Spearman’s correlation (r_s_) evaluated the significance and strength of relationships between dependent and independent variables. Forward stepwise multivariate regression included only significant predictors: age, time since SCI, sex, marital status, level and extent of SCI, cause (traumatic vs. non-traumatic), and ARC participation (new vs. recurrent). The interpretation of the model’s determination coefficient fit according to Falk and Miller was used (R^2^ ≥ 10%). Significance was set at *p* < 0.05. All analyses were conducted using the Statistica data analysis software system.

## 3. Results

Of 177 individuals with SCI across 16 consecutive ARCs, 125 met the inclusion criteria and were included in this study (Figure 1). Most exclusions were due to a different diagnosis and inability to understand and respond to written questions. Among participants, 83 (66%) were male, 79 (63%) had paraplegia, and 71 (57%) had incomplete lesions. The mean age was 39.7 years (SD = 15.2), and the mean time since injury was 8.0 years (SD = 8.9). Eighty one (65%) were first-time ARC participants, 95 (75%) had at least secondary education, 15 (12%) were employed, and 10 (8%) were students (Table 1).

Mobility outcomes showed marked improvements across several domains, with four of five components improving significantly between T1 and T2 (Table 2). The QEWS score increased from 11.8 to 14.4 (*p* < 0.001; d = 2.3), and individually, 95 (76%) participants achieved MCID (difference ≥ 1 point). WST-Q capacity increased from 48.8 to 64.1 (*p* < 0.001; d = 2.2), and individually, 95 (76%) participants achieved MCID (difference ≥ 5 point) between T1/T2 and 69 (55%) between T1/T3. WST-Q confidence increased from 50.7 to 70.4 (*p* < 0.001; d = 1.6), and individually, 79 (63%) participants achieved MCID (difference ≥ 6 point) between T1/T2 and 60 (48%) between T1/T3. The SCIM-SR indoor/outdoor mobility domain increased from 6.1 to 7.1 (*p* = 0.001; d = 0.7), and individually 46 (48%) achieved MCID (difference ≥ 1 point) between T1/T2. MSES Item 5 (“Get out of my house”) showed no change. Three months later (T3) the gains in WST-Q capacity (*p* < 0.001; d = 1.2) and confidence (*p* = 0.001; d = 0.7) persisted, whereas SCIM-SR indoor/outdoor mobility and MSES Item 5 didn’t change. Overall, the Mobility Index rose by 10.5% from T1 to T2 (Figure 2a), and by 8.8% from T1 and T3 (Figure 2b).

Improvements were also observed in bowel and bladder function, though to a lesser extent. Between T1 and T2 (Table 2), SCIM-SR bowel function increased from 6.6 to 8.2 (*p* = 0.001; d = 0.7) and toilet use from 2.6 to 3.1 (*p* = 0.015; d = 0.5), whereas SCIM-SR bladder and MSES Item 2 (“Avoid bowel accidents”) showed no change. None of these improvements were retained at T3. Overall, the Bowel and Bladder Index rose by 13.2% from T1 to T2 (Figure 3), but gains were not sustained at T3.

Changes in sexual function followed a different pattern. Between T1 and T2, MSES Item 6 (“Satisfying sexual relationship”) increased from 3.6 to 4.3 (*p* = 0.004; d = 0.6) (Table 2) and this gain persisted at T3 (*p* = 0.004; d = 0.5). In addition, LiSat-11 sexual life improved at T3 (*p* = 0.010; d = 0.6). Correspondingly, the Sexuality Index increased by 11.8% from T1 to T2 (Figure 4a), and by 10.5% from T1 to T3 (Figure 4b).

Similar trends were found for hand and arm function (Table 2). By T2, SCIM-SR self-care domain increased from 9.7 to 14.8 (*p* < 0.001; d = 0.9), and individually, 33 (72%) participants achieved MCID (difference ≥ 1 point) between T1/T2 and 34 (74%) between T1/T3. The MSES Item 1 (“Maintain personal hygiene”) from 4.7 to 6.1 (*p* = 0.001; d = 0.6). At T3, these improvements were sustained: both SCIM-SR self-care (*p* < 0.001; d = 0.9) and LiSat-11 self-care (*p* < 0.001; d = 0.9) remained higher than the baseline, while MSES Item 1 continued to show a moderate improvement (*p* = 0.003; d = 0.6). Overall, the Hand and Arm Index rose by 24.6% at T2 (Figure 5a) and by 22.8% at T3 compared to T1 (Figure 5b).

To explore factors associated with functional gains, correlations were examined between selected independent variables and the Mobility, Bowel and Bladder, Sexual, and Hand and Arm indices at both follow-up points (Appendix A). Based on these correlations, two separate forward stepwise multivariate regression models were built for T2 − T1 differences. In the first model, previous ARC attendance and cause of SCI together explained 13% of the variance in Mobility Index change (R^2^ = 0.13; F(2,93) = 6.92; *p* = 0.002). Participants attending ARC for the first time and those with traumatic SCI demonstrated greater improvements than recurrent attendees and those with non-traumatic causes. In the second model, sex explained 13% of the variance in Hand and Arm Index change (R^2^ = 0.13; F(1,42) = 6.17; *p* = 0.017), with males with tetraplegia showing greater gains than females. None of the tested variables significantly explained variance in other functional outcomes.

## 4. Discussion

Our study underscores the value of peer-led ARC in addressing key health priorities among community-dwelling individuals with SCI. Overall, individuals with SCI who participated in ARC in Poland achieved the greatest gains in hand and arm function, followed by improvements in sexual function, mobility, and lastly bowel and bladder function. With few exceptions, the predictors examined did not account for these outcomes showing their weak overall associations with demographic and injury characteristics. Factors that predict gains in health priorities during ARC warrant further investigation.

Participants with tetraplegia showed significant improvements in hand and arm function during the ARC, and these gains were retained at the 3-month follow-up. Improvements included aspects of self-care and hygiene, all constituting core ADL. During ARC, participants practice washing and dressing several times per day with guidance from peer mentors, naturally integrated into daily schedule such as morning and evening routines and activities such as swimming. Participants with tetraplegia were continuously encouraged and supported during ARC to perform ADL independently. Greater independence in self-care reduces reliance on personal assistants and may be perceived as directly meaningful in everyday life. This combination of frequent practice, immediate functional payoff and positive feedback from peer mentors is likely to have strengthened intrinsic motivation and habit formation, which may explain why gains in hand and arm function were maintained at three months. While self-care is typically emphasized during in-patient rehabilitation, the substantial late-phase gains observed here suggest both the unique impact of peer support and participants’ readiness for change. As many persons with tetraplegia rely on costly personal assistants for ADL, these improvements may translate into decreased dependence on external daily help.

After participating in the camp, participants reported greater confidence in their ability to have a satisfying sexual relationship, and at the 3-month follow-up they reported significantly higher satisfaction with their sexual life compared with baseline. These findings are partly consistent with findings from the Swedish Inter-PEER study, in which participants improved their confidence in their ability to have a satisfying sexual relationship at camp completion, but these gains were not sustained at 3 months [13]. These outcomes likely reflect the ARC program’s dedicated educational sessions on sexuality and fertility, as well as the many informal peer discussions where personal experiences such as parenting are shared. It is possible that many participants were systematically introduced to sexuality after SCI for the first time during ARC. Within the dedicated educational session, participants receive information about building and maintaining partnership relations following SCI, as well as about medical and rehabilitative options to enhance sexual functioning. Combined with informal peer discussions and role modelling, this may increase self-efficacy, thereby fostering hope that a satisfying relationship and sexual life after SCI is attainable. Nevertheless, sexual life remained the least satisfying domain in our cohort, aligning with existing literature. More specifically, across European (Poland, Sweden, the Netherlands, England) and Asian (China, India, Vietnam, Sri Lanka) studies, mean sexual-satisfaction scores range only from 1.5 to 3.3 on a 0–6 scale [5,23]. Sexuality is rarely addressed in in-patient rehabilitation [24], yet evidence from other conditions highlights the added value of peer support in this area [25]. Greater attention to sexuality—introduced early, timed appropriately, and delivered by diverse informants—could improve satisfaction with this often-neglected topic.

Participants achieved substantial improvements in wheelchair skill capacity and confidence at ARC completion, maintained at follow-up. However, confidence in getting outside their house did not change. Mastering specific wheelchair skills may be an essential first step, while navigating unpredictable outdoor environments is inherently more complex. Our results confirm that wheelchair skills training is a central pillars of the ARC program [8]. Baseline QEWS score in this cohort averaged 13.1, yet participants achieved a 3.1-point gain—greater than reported in ARC studies from Poland [12], Sweden [13], Botswana [11] and Morocco [26]. A comparable 3.1-point improvement was observed in an Australian 10-week program involving 100 individuals with SCI (baseline QEWS: 10 for tetraplegia, 17 for paraplegia) [14]. These findings highlight the efficiency of ARC in producing rapid, sustained mobility gains and underscore the need for continued efforts to build participants’ confidence and ability to handle the challenges of outdoor mobility—skills essential for social inclusion and participation [27].

Our findings show that participants achieved the second largest gains in bowel and bladder function, reflected in reported improvements in toilet use, toilet transfers, and bladder management at ARC completion. These gains were not retained at the 3-month follow-up. Findings from the Swedish Inter-PEER study demonstrated sustained gains in bowel management in both capacity and self-efficacy [13]. During the ARC, topics such as urinary tract infection prevention and bowel management are addressed through group discussions and informal peer interactions, where personal strategies for bladder and bowel care are shared. During ARC there is limited scope to fundamentally change an individual’s bowel or bladder regime, which often requires medical review, specialist equipment and coordination with caregivers. The camp therefore focuses primarily on promoting independence in practical aspects such as toileting, use of suppositories and independent handling of urological supplies (intermittent catheter or condom drainage). The short-term gains observed in these areas were not sustained at three months, possibly because participants returned to home environments where care routines are shared with family members or assistants, bathroom accessibility is suboptimal, or opportunities to practice independent management are restricted. These structural and contextual constraints may limit the consolidation of new bowel and bladder behaviors outside the camp setting. Because bowel and bladder management needs can change over time, these ADL should continue to be monitored during the face-to-face training sessions with peer mentors in participants’ homes after camp. A 20-year longitudinal study found that over half of individuals living with SCI for over 40 years required changes to their bladder or bowel management for medical or practical reasons [28]. Our findings add to existing evidence that bladder management can improve during ARC, complementing standard inpatient or outpatient rehabilitation [29].

### Study Strengths and Limitations

A key strength of this study was the harmonization of items, domains, and outcome measures across health priorities. We aligned outcomes with those of the broader Inter-PEER project and applied a predefined three-stage selection process. For some priorities (e.g., sexual function), available measures were limited to two single items from different instruments, which may restrict the level of detail in those analyses. Certain measures—such as using self-care to represent hand/arm function—also capture related abilities (e.g., trunk control), introducing some construct overlap rather than systematic bias.

Combining complementary constructs (activity, function, satisfaction, self-efficacy) provides a broad view of change, recognizing that individuals with SCI may progress in some areas but not others. Incremental change is meaningful within longer-term behavior change [30], and was therefore monitored and reported. Findings should be read with these measurement considerations in mind. Finally, the sample was heterogeneous in epidemiological and demographic characteristics, including 17% with spina bifida—generally younger and with longer duration of disability than participants with traumatic SCI. This variability should be considered when generalizing the results. The findings are most applicable to community-dwelling individuals with SCI in settings with limited post-acute and community rehabilitation services, such as Poland, and may differ in health systems with more comprehensive rehabilitation pathways or broader access to peer-led programs.

## 5. Conclusions

Participation in peer-led ARC in Poland was associated with meaningful short-term improvements in several priority health domains for people with SCI, particularly hand and arm function, sexual well-being, wheelchair skills, and bowel and bladder management. Some of these gains, including enhanced self-care and mobility, were sustained at three months. The findings underscore the unique value of peer mentorship, structured skill training, and open discussion of sensitive topics such as sexuality—areas often underemphasized in standard inpatient or outpatient rehabilitation. Incorporating peer-led, community-based programs like ARC into the broader continuum of SCI care may help reduce long-term dependence on personal assistance and promote greater independence and social participation.

## Figures and Tables

**Figure 1 jcm-15-00176-f001:**
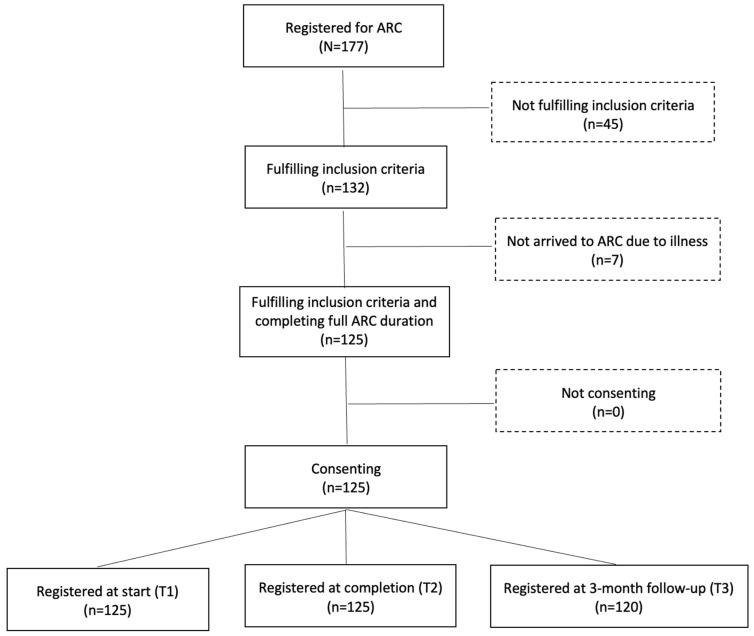
Number of participants at different stages of the study.

**Figure 2 jcm-15-00176-f002:**
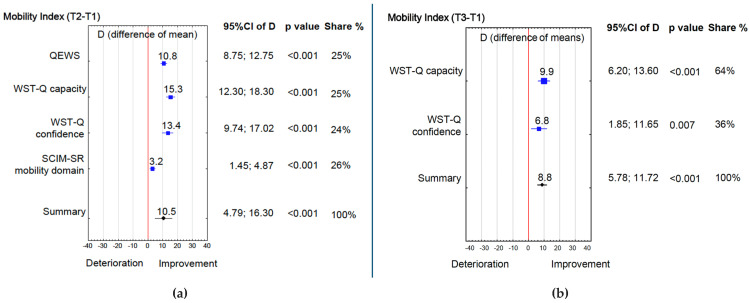
(**a**) Mobility Index (T2 − T1); (**b**) Mobility Index (T3 − T1). 95%CI of D—95% confidence interval of maximum score difference possible to obtain in a given scale, T1, T2, T3—time of measurement, Colors: Red—significance limit, Blue—Index element, Black—Index summary.

**Figure 3 jcm-15-00176-f003:**
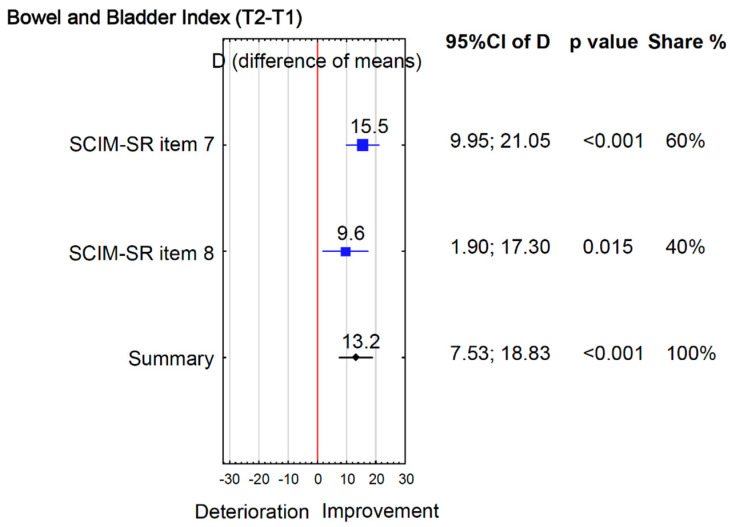
Bowel and Bladder Index (T2 − T1).

**Figure 4 jcm-15-00176-f004:**
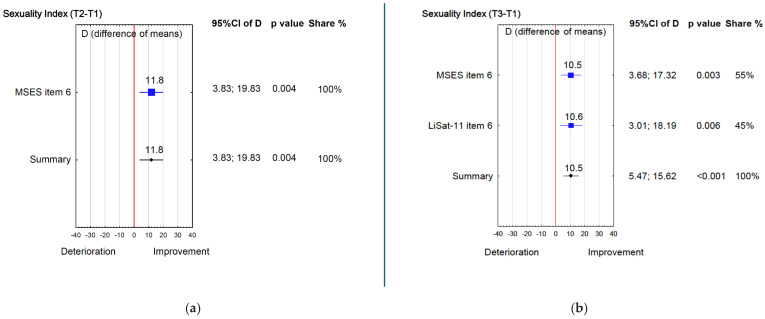
(**a**) Sexuality Index (T2 − T1); (**b**) Sexuality Index (T3 − T1).

**Figure 5 jcm-15-00176-f005:**
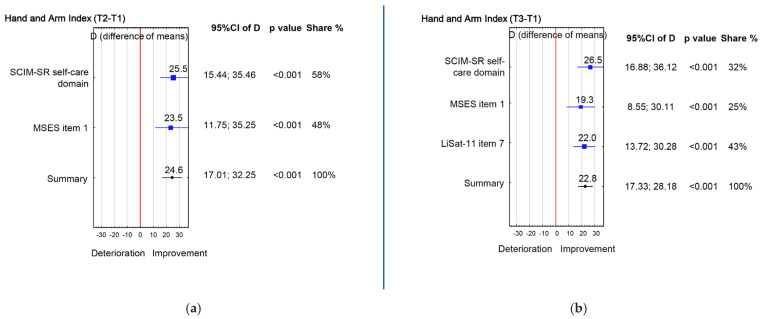
(**a**) Hand and Arm Index (T2 − T1); (**b**) Hand and Arm Index (T3 − T1).

**Table 1 jcm-15-00176-t001:** Demographic and injury characteristics of 125 study participants with SCI.

Demographic and Injury Characteristics	Participants with SCI (N = 125)
Sex (n, %)	
Male	83 (66.4)
Female	42 (33.6)
Age for those with TSCI (mean years ± SD)	42.2 ± 14.2
Age for those with NTSCI (mean years ± SD)	32.3 ± 15.8
Marital status (n, %)	
Single	59 (47.2)
Married	50 (40.0)
Cohabiting or in a partnership	7 (5.6)
Separated or divorced	7 (5.6)
Widowed	2 (1.6)
Education (n, %)	
Primary	27 (20.8)
Vocational	35 (28.0)
Secondary	33 (26.4)
Post-secondary	4 (3.2)
Bachelor	9 (7.2)
Post-graduate	18 (14.4)
Employment status (n, %)	
Employed	15 (12.0)
Not employed	38 (30.4)
Student	10 (8.0)
Retired due to health condition	51 (40.8)
Retired due to age	11 (8.8)
Level o SCI (n, %)	
Paraplegia	79 (63.2)
Tetraplegia	46 (36.8)
Completeness of SCI (n, %)	
Complete	54 (43.2)
Incomplete	71 (56.8)
Cause of traumatic SCI (n, %)	93 (74.4)
Sport	12 (9.6)
Recreation	13 (10.4)
Work related	10 (8.0)
Traffic accident	28 (22.4)
Fall < 1 m	4 (3.2)
Fall > 1 m	26 (20.8)
Cause of non-traumatic SCI (n, %)	32 (25.6)
Spina bifida	21 (16.8)
Degenerative changes	2 (1.6)
Tumour benign	6 (4.8)
Tumour malignant	1 (0.8)
Vascular disorders	1 (0.8)
Infection	1 (0.8)
Time since injury for those with TSCI (mean years ± SD)	5.7 ± 7.2
Time since disease for those with NTSCI (mean years ± SD)	14.8 ± 10.2
Attendance in Active Rehabilitation Camps (n, %)	
First comers	81 (64.8)
Recurrent comers	44 (35.2)

Note: TSCI: traumatic spinal cord injury; NTSCI: non-traumatic spinal cord injury.

**Table 2 jcm-15-00176-t002:** Study findings matched to health priorities following SCI.

						T2 − T1		T3 − T1	
Health Priorities Following SCI	n	Range	T1	T2	T3	*p* Value	Effect Size	*p* Value	Effect Size
Mobility (walking & wheeling) ^1^									
QEWS (total score)	96	1–25	11.8 ± 7.2	14.4 ± 7.4	N/A	**≤0.001**	2.3	N/A	N/A
WST-Q capacity (total score)	96	1–100	48.8 ± 25.3	64.1 ± 25.5	58.7 ± 28.2	**≤0.001**	2.2	**≤0.001**	1.2
WST-Q confidence (total score)	96	1–100	57.0 ± 27.9	70.4 ± 25.1	63.6 ± 28.1	**≤0.001**	1.6	**0.001**	0.7
SCIM-SR mobility indoors/outdoors (domain score)	96	0–30	6.1 ± 2.5	7.1 ± 2.0	6.7 ± 2.4	**0.001**	0.7	0.144	N/A
MSES item 5. Get out of my house	96	1–7	4.9 ± 2.0	5.3 ± 1.8	4.9 ± 1.9	0.147	N/A	0.781	N/A
Bowel and bladder function ^1^									
SCIM-SR item 6. Bladder	116	0–15	4.6 ± 4.1	5.3 ± 4.2	5.0 ± 4.3	0.111	N/A	0.428	N/A
SCIM-SR item 7. Bowel	116	0–10	6.6 ± 4.7	8.2 ± 4.1	7.4 ± 4.4	**0.001**	0.7	0.228	N/A
SCIM-SR item 8. Toilet use	116	0–5	2.6 ± 1.9	3.1 ± 1.8	2.7 ± 1.8	**0.015**	0.5	0.751	N/A
MSES item 2. Avoid bowel accidents	116	1–7	4.6 ± 1.8	4.9 ± 1.7	4.7 ± 1.8	0.188	N/A	0.427	N/A
Sexual function ^1^									
MSES item 6. Satisfying sexual relation	99	1–7	3.6 ± 1.9	4.3 ± 1.8	4.2 ± 1.8	**0.004**	0.6	**0.004**	0.5
LiSat-11 item 6. Sexual life	99	1–6	2.5 ± 1.7	N/A	3.1 ± 1.7	N/A	N/A	**0.010**	0.6
Arm and hand function ^2^									
SCIM-SR self-care (domain score)	44	0–20	9.7 ± 5.8	14.8 ± 5.5	15.0 ± 5.1	**≤0.001**	0.9	**≤0.001**	0.9
MSES item 1. Maintain personal hygiene	44	1–7	4.7 ± 2.1	6.1 ± 1.3	5.9 ± 1.5	**0.001**	0.6	**0.003**	0.6
LiSat-11 item 7. Manage self-care	44	1–6	3.1 ± 1.5	N/A	4.4 ± 1.2	N/A	N/A	**≤0.001**	0.9

Note: ^1^ All study participants; ^2^ Only participants with tetraplegia; T1, T2, T3—time of measurement; Bold—significant result at *p* < 0.05; N/A—not applicable.

## Data Availability

Aggregated data are available upon reasonable requests to the principal investigator.

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
