# Peer review of "Health Priorities and Participation in Peer-Led Active Rehabilitation Camps Among Persons with Spinal Cord Injury: A Prospective Cohort Study"

_jcm, 2025, doi:10.3390/jcm15010176_

Round 1

Reviewer 1 Report

Comments and Suggestions for Authors

Health Priorities and Participation in Peer-Led Active Rehabilitation Camps Among Persons with Spinal Cord Injury: A Prospective Cohort Study

Journal of  Clinical Medicine

REVIEW THE MANUSCRIPT 

Methods

Page 2, line 83:  Can you briefly explain the Inter-PEER project?

Page 2, Line 84: Specify the precise dates of the patients included in the study, including months.

Page 2, line 84-85: Provide the number of the Bioethics Committee's approval. Consider providing a copy of Local Ethical Committee consent document required for research involving interaction with patients (e.g. as supplementary material).

Page 2, line 88-92: Did all the participants undergo early rehabilitation in hospital after SCI?

Where did the patients come from (location)? Please clarify, did the participants represent different parts of your country? 

Please add exclusion criteria.

Page 3, line: 94-106: Can you describe how long each training session lasted: the ADL programme, educational workshops, etc.?

Page 3, Lines 90–92: Expand on the information that patients signed an informed consent form to participate in the study: i.e., was participation voluntary and could anyone withdraw at any time?

It would be useful to include the STROBE Statement for Cohort Study. Consider including STROBE checklist, e.g. as supplementary material.

Figure 1: Please indicate the exclusion criteria and the reasons for participants dropping out of the study (T3).

Was the selection error calculated? Please consider including such calculations.

Results

Explain all abbreviations in the table 1 and table 2.

Page 5, Lines 217-222: Avoid repeating information in the main text that is already included in the tables.

Page5: line 224-229: Consider transferring this part of results to the methodology section.

Add explanatory captions to the figures, including all abbreviations used in them.

The figures are not included in the main text and are difficult to review. I suggest that in the second review, they be included in the main text. The figures lack captions and explanations of abbreviations.

Author Response

Response to Reviewer 1 comments are attached as a WORD file.

Reviewer 2 Report

Comments and Suggestions for Authors

Major comments:
1. The manuscript introduces four indices (Mobility, Bowel & Bladder, Sexuality, Hand and Arm) integrating different items and domains. This strategy is intriguing, yet it is regrettable that there is an absence of an in-depth discussion of its conceptual and psychometric validity.

  • It is recommended that justification be provided for the combination of the selected instruments into a single metric.
  • A discussion is required on the evaluation of redundancy or collinearity between components.
  • It is imperative that an explanation is provided to demonstrate why factor analysis or a more sophisticated weighting method was not utilised.

2. The application of imputation for data sets with ≤20% missing values necessitates further elucidation. This approach may be subject to bias, stemming from the assumption of intra-individual consistency.

  • It is imperative to substantiate the selection of this approach in comparison to alternative methodologies, such as multiple imputation.
  • It is recommended to incorporate a sensitivity analysis to ascertain whether the results vary in the absence of imputation.

3. Regression models explain only approximately 13% of the variability in two indices, and none in the remaining ones.

  • A discussion is warranted concerning whether this outcome reflects limitations of the model, an insufficient number of predictors, or the multifactorial nature of the rehabilitation process.
  • However, the available information on assumption verification (homoscedasticity, collinearity, autocorrelation) is inadequate.
  • It would be beneficial to provide a rationale for the exclusion of clinically relevant interactions.

4. The manuscript reports statistical significance; however, it does not interpret the changes with respect to minimum clinically important difference (MCID) thresholds, when available.

  • This is a prerequisite for evaluating the practical relevance of the programme.
  • It is recommended that the reader clarify whether there are published MCIDs for QEWS, WST-Q, SCIM-SR, or LiSat-11.

5. The discussion could be expanded to explain why some domains (sexuality, upper limb) maintain improvements at three months, while others (bowel/bladder) do not. In order to enhance the quality of the ensuing discussion, it would be beneficial to incorporate a more in-depth clinical and behavioural interpretation.

Minor comments:

  • The participant selection process is adequately explained, but it is suggested that a flow chart be included within the main manuscript.
  • The internal consistency values of the WST-Q (α > 0.95) could be indicative of redundancy. A suitable location for this comment would be the section entitled 'Limitations'.
  • The incorporation of confidence intervals within composite indices would enhance analytical transparency.
  • The description of the ARC protocol could be condensed to improve the flow of the methods section.
  • It is recommended that the variability in diagnoses (e.g., percentage with spina bifida) be briefly mentioned as a potential factor that could affect the generalisation of the results.
  • The translation was rendered using the DeepL.com machine translation service (free version).

Author Response

Response to Reviewer 2 comments are attached as a WORD file.

Reviewer 3 Report

Comments and Suggestions for Authors

This manuscript offers a well-designed and clearly written prospective cohort study examining whether peer-led Active Rehabilitation Camps (ARC) help individuals with spinal cord injury improve in domains they commonly identify as top health priorities, however several methodological and reporting issues limit the strength of the conclusions.

The construction of the four composite “priority indices” needs fuller justification, because the indices combine conceptually different constructs—such as performance-based mobility tasks, self-efficacy items, and satisfaction ratings—and it is not entirely clear why these heterogeneous measures should be aggregated or how this affects interpretability, especially when the indices are later used as dependent variables in regression models. Related to this, the handling of missing data is insufficiently described: ipsative imputation is an uncommon choice for multi-item validated instruments, and the manuscript does not specify how frequently imputation was applied or whether results changed when only complete cases were analyzed, making it difficult to assess the robustness of the reported effect sizes.

The two regression models that reached significance explained only 13% of the variance each, which is relatively modest, and yet the narrative occasionally implies stronger predictive value than the data support; these interpretations would benefit from more cautious language emphasizing the weak overall associations with demographic and injury characteristics.

The cohort itself is highly heterogeneous—with wide variation in age, time since injury, lesion completeness, and inclusion of 17% of participants with spina bifida—and the analysis does not adequately account for potential differences between these subgroups, nor does it provide baseline comparisons for first-time versus recurrent ARC attendees, who likely differ in functional level and expectations.

Although the findings on bowel and bladder management show improvements at camp completion, these gains were not sustained at the three-month follow-up, and the discussion would be stronger if it more directly addressed the transient nature of these changes rather than framing them mainly as confirmation of program benefit.

Despite these limitations, the study has strengths, including the use of validated SCI-specific measures, detailed documentation of intervention fidelity, a sample size meeting protocol requirements, and clear alignment with the broader Inter-PEER initiative. With clearer reporting of index construction, more transparency around missing data and subgroup differences, and a more measured interpretation of the regression findings and transient outcomes, the manuscript would be substantially strengthened.

Author Response

Response to Reviewer 3 comments are attached as a WORD file.

Round 2

Reviewer 2 Report

Comments and Suggestions for Authors

Thamks you to provide a thorough and appropriate response to the comments raised during the first review round. The revised version (V2) shows substantial improvements in methodological clarity, statistical transparency, and conceptual coherence between health priorities, outcome measures, and analytical strategies.

In particular, the clarification regarding the exploratory nature of the composite indices, the inclusion of sensitivity analyses, the explicit integration of MCID thresholds, and the expansion of the clinical discussion significantly strengthen the manuscript. Overall, the study now provides a solid and clinically relevant contribution to the literature on peer-led community-based rehabilitation for persons with spinal cord injury

Reviewer 3 Report

Comments and Suggestions for Authors

The authors adequately responded to all comments/concerns.